# 3-D TEE Mitral Valve Segmentation and Mesh Reconstruction with Real-Time Quality Assurance

Phat K. Huynh[1], Jacques Kpodonu[2], Minh Huu Nhat Le[3], Dang Nguyen[4], Phi Huynh[5], Thuan Q. Phan[6], Olabiyi H. Olaniran[4], Tam Tran[7], Heath Rutledge-Jukes[8], Yanwen Xu[9], and Dinh H. Nguyen[10]†

[1]Department of Industrial and Systems Engineering, North Carolina A&T State University, Greensboro, NC, USA
[2]Division of Cardiac Surgery, Beth Israel Deaconess Medical Center, Boston, MA, USA
[3]Department of Interventional Cardiology, Methodist Hospital, Merrillville, IN, USA
[4]Harvard T.H. Chan School of Public Health, Harvard University, Boston, MA, USA
[5]PASSIO Laboratory, North Carolina A&T State University, Greensboro, NC, USA
[6]Department of Cardiovascular Surgery, University Medical Center, Ho Chi Minh City, Vietnam
[7]John T. Milliken Department of Medicine, Washington University in St. Louis School of Medicine, St. Louis, MO, USA
[8]Washington University in St. Louis School of Medicine, Centaur Labs Inc, King of the Curve LLC, Saint Louis, MO, USA
[9]Department of Mechanical Engineering, University of Texas at Dallas, Richardson, TX, USA
[10]University of Medicine and Pharmacy at Ho Chi Minh City, Ho Chi Minh City, Vietnam
†Corresponding author: dinh.nh@umc.edu.vn

*Abstract*—Accurate, real-time segmentation of the mitral valve (MV) from three-dimensional transesophageal echocardiography (3-D TEE) remains a technical bottleneck for intra-procedural guidance and patient-specific modeling. We present a vision-transformer based pipeline that performs voxel-wise MV segmentation, entropy-driven quality assurance (QA), and instantaneous mesh export on the public MVSeg2023 dataset. A Swin-UNETR backbone was trained on 105 training and 30 validation volumes (voxel size 0.6 mm). The model employs temperature-scaled logits and computes per-voxel Shannon entropy to flag uncertain predictions. Leaflet meshes are generated on-the-fly via marching cubes and Taubin smoothing. On 40 held-out test volumes the method achieved a Dice coefficient of $0.83 \pm 0.05$ and a 95th-percentile Hausdorff distance of 4.2 mm. 100 % cases passed the QA gate (Entropy$_{95}$ < 0.80), with no significant correlation between entropy and Dice ($r = 0.016$, $p = 0.92$). End-to-end inference, including meshing, averaged 104 ms per volume on a single NVIDIA A100 GPU while peaking at 4.0 GiB of video memory. The proposed echo-only transformer pipeline delivers state-of-the-art accuracy, built-in reliability estimates, and real-time performance, satisfying key clinical usability requirements. To our knowledge this is the first 3-D TEE MV segmentation framework that unites transformer representations, uncertainty-aware QA, and mesh reconstruction within 0.1 s, paving the way for routine intra-procedural deployment.

*Index Terms*—Mitral valve, transesophageal echocardiography, image segmentation, vision transformers, quality assurance, mesh reconstruction

## I. INTRODUCTION

Valvular heart disease (VHD) remains a significant clinical and economic burden worldwide. In the United States alone, Medicare data document more than 100,000 open or minimally invasive mitral valve (MV) procedures annually, with in-hospital mortality approaching 5% [1]. Epidemiologic reviews indicate that clinically relevant mitral regurgitation or stenosis affects almost 3% of adults and more than 10% of those over 65, underscoring the pressing need for rapid, objective morphological assessment at the point of care.

Currently, three-dimensional trans-esophageal echocardiography (3-D TEE) is the reference imaging modality for intra-operative guidance because it delivers isotropic, high-frame-rate volumes that can be rendered from the "surgeon's view," thereby facilitating real-time interpretation of leaflet morphology, annular dynamics, and sub-valvular apparatus without ionizing radiation [2], [3]. Multiple prospective series have demonstrated that 3-D TEE alters surgical decision-making in up to one-third of mitral cases and improves post-repair durability relative to conventional 2-D approaches [2], [4]. Nonetheless, manual delineation of the anterior (A) and posterior (P) leaflets typically consumes 10–15 min per volume and exhibits substantial inter-observer variability, limiting its practicality for time-critical applications such as trans-catheter edge-to-edge repair or robotic assistance [2], [5].

Over the past two decades, automated MV segmentation from 3-D TEE has progressed through three successive waves. The first wave attempted to "transfer knowledge" from expertly labeled exemplars to unseen studies. Multi-atlas label fusion combined with medial-axis or spline-based shape priors delivered the first fully automatic leaflet masks. Canonical examples include the probabilistic atlas of *Pouch et al.* [6], which achieved sub-millimeter annular localization; and the 4-D statistical model of *Ionasec et al.* [7], which distilled thousands of annotated frames into a deformable template for beat-to-beat tracking. Other variants employed J-spline or B-spline surfaces, level-set evolution, or non-negative matrix factorization; Dice coefficients typically fell below 0.70 and interactive correction was frequently necessary [6], [8], [9].

Second-generation methods employ fully convolutional networks (FCNs), and the introduction of GPU-accelerated deep learning has shortened inference latency by roughly two orders of magnitude while raising segmentation accuracy to clinically acceptable levels. DeepMitral—the first 3-D FCN for TEE—attained a Dice of 0.81 with roughly 1s inference on

a high-end GPU [10]. Subsequent refinements added dilated kernels, squeeze-and-excitation attention, or landmark-aware post-processing, but performance has plateaued at 0.83 Dice [4]. To curb annotation costs, semi-supervised teacher–student paradigms now exploit pseudo-labels from unlabeled 4-D sequences, halving expert workload while preserving accuracy [11]. Recently, *Chen et al.* [12] introduced a two-stage pre-training strategy in which a 3-D CNN first learns to classify systolic versus diastolic frames before being fine-tuned for leaflet segmentation. Nevertheless, classical FCNs possess limited receptive fields, struggle with the thin edges of the leaflets, and offer no quantification of prediction reliability.

The third generation of automated MV segmentation is defined by transformer architectures and geometric deep-learning approaches. Motivated by the success of vision transformers in natural imagery, hybrid encoder–decoder networks—such as CoST-UNet and MCCT-UNet—embed Swin-Transformer blocks to capture global context, achieving state-of-the-art results for ventricular and atrial structures across CT, MR, and ultrasound modalities [13], [14]. Within the MV domain, MV-graph neural network (GNN) maps voxel features to temporally continuous surface meshes via graph neural decoders, facilitating valve-in-valve simulations but still relying on an accurate initial voxel mask [15], [16]. End-to-end voxel-to-mesh generators such as HybridVNet further hint at bypassing voxel masks altogether, although these remain experimental for cardiac anatomy [17]. Recently, *Wifstad et al.* introduced an attention-gated UNet with deep supervision on 2-D transthoracic views while simultaneously localizing annular hinge points [18]. Real-time performance, case-level uncertainty estimates, and one-click mesh export have yet to be delivered in a single cohesive framework.

Quality assurance (QA) has emerged as a non-negotiable prerequisite for the safe clinical deployment of deep-learning segmentation systems. Recent reviews emphasize that flagging potentially unreliable outputs is now considered as important as maximizing mean Dice scores [19], [20]. Uncertainty can be estimated in several ways. Classical deep ensembles compute the variance across independently trained networks, yielding well-calibrated confidence maps but multiplying training and inference cost by the ensemble size [20]. Monte-Carlo (MC) dropout performs stochastic weight masking at test time to approximate the posterior and has been adopted in U-Net variants such as MCU-Net and Bayesian U-Net, yet still requires 20–50 forward passes per study [21]. Test time augmentation (TTA) instead samples plausible image transformations; although mathematically elegant, typical implementations run 8–16 augmentations and thus incur comparable overhead [22]. Single-pass approaches have therefore gained traction. Layer-ensembles, evidential deep learning, and one-shot surrogates reduce the burden to a single evaluation but have only recently matched the calibration of MC methods [23].

Among one-shot metrics, Shannon entropy of the softmax logits is particularly attractive because it is architecture-agnostic and incurs negligible extra compute. Multiple independent studies have demonstrated a strong, monotonic correlation between voxel-wise or case-level entropy and Dice error in CT liver and prostate benchmarks, cardiac MRI, and dermoscopic lesion segmentation [24], [25]. Entropy maps have also been leveraged to drive human-in-the-loop annotation and matting of ambiguous lesion boundaries [26]. Despite this evidence, no prior work has embedded entropy-based QA inside a real-time 3-D TEE MV pipeline, as existing mitral segmentation studies either omit uncertainty altogether [10] or rely on computationally intensive MC strategies [27].

The present study addresses those challenges through a methodology that unites segmentation, quality assurance, and mesh reconstruction for 3-D TEE imaging of the MV. The specific contributions of our study are: (i) we adapt the Swin-UNETR model [28] to the MVSeg 2023 benchmark, establishing the first transformer reference on this public dataset; (ii) A 95th-percentile Shannon entropy metric identifies unreliable cases without ensemble overhead; and (iii) GPU-accelerated marching cubes and Taubin smoothing [29] yield watertight STL meshes in 104 ms per volume.

## II. METHODS

### A. Dataset

For training and evaluation, we employ the MVSeg 2023 dataset [30], the first openly-licensed 3-D TEE dataset with voxel-level mitral-leaflet annotations. In this dataset, all scans are acquired on a Philips EPIQ cardiac ultrasound platform using mid-oesophageal, en-face TEE probes under routine clinical standards at King's College Hospital, London. Raw Philips DICOM volumes are converted to Cartesian grids with Philips QLab and exported as anonymized NIfTI files. The dataset comprises 150 single-frame end-diastolic volumes (one per subject) partitioned into 105 training, 30 validation, and 40 test studies. Every voxel is labeled as background (0), posterior leaflet (1), or anterior leaflet (2).

### B. Overview

The proposed pipeline converts a single-frame, 3-D TEE volume into both a voxel-level leaflet mask and a watertight STL surface (Fig. 1). It has four stages: *(i) pre-processing* normalizes intensity, enforces isotropic spacing, and crops the foreground; *(ii) segmentation* applies a lightweight Swin-UNETR network [28] to produce posterior–anterior leaflet logits; *(iii) quality assurance* derives a 95th-percentile Shannon-entropy score from the same forward pass, flagging uncertain cases without Monte-Carlo overhead; and *(iv) surface reconstruction* executes GPU-accelerated marching cubes followed by Taubin smoothing [29], yielding an STL mesh that is ready for computational modeling or intra-operative display.

### C. Pre-processing

All experiments start from the single–frame NIfTI volumes provided by MVSeg 2023. A pre-processing chain, implemented in `MONAI 1.3`, brings scans onto a common spatial and intensity grid before it is presented to Swin-UNETR.

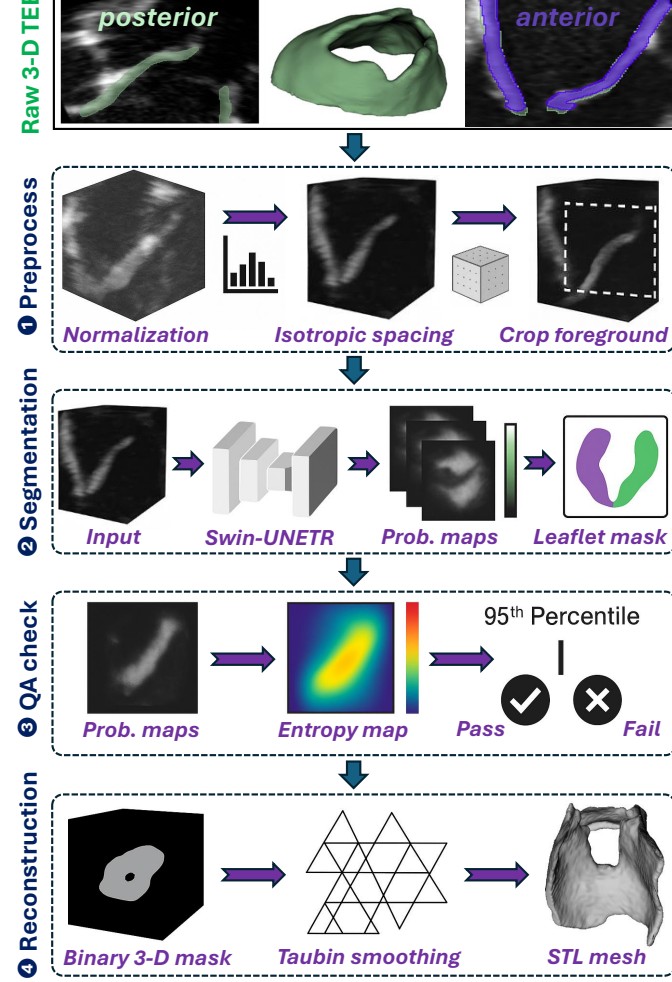

Fig. 1. Overview of the proposed four-stage pipeline.

*1) Intensity normalization:* For each volume, we compute the $1^{\text{st}}$ and $99^{\text{th}}$ percentile intensities, $p_1$ and $p_{99}$, and linearly rescale the voxel values to the interval $[0, 1]$:

$$I_{\text{norm}} = \text{clip}\left(\frac{I_{\text{raw}} - p_1}{p_{99} - p_1}, 0, 1\right) \qquad (1)$$

This operation removes scanner-specific gain and suppresses rare, extreme echoes.

*2) Isotropic resampling:* The normalized image and its binary label map are both resampled to an isotropic voxel size of 0.6 mm using third-order B-spline interpolation [31] for the image and nearest-neighbor interpolation for the labels.

*3) Foreground cropping:* We compute the tightest axis-aligned bounding box around the leaflet label and extend it by five voxels in each direction to guarantee spatial context. The resulting sub-volume is zero-padded so that each spatial dimension is a multiple of 32; this is required by the shifted-window mechanism in Swin-UNETR.

### D. Network Architecture

The proposed backbone is Swin-UNETR [28] —a 3-D extension of the Swin Transformer that embeds four hierarchical self-attention stages inside a U-Net–style encoder–decoder. A layer-by-layer specification is given in Table I.

*1) Patch embedding and encoder:* The input tensor $\mathbf{x} \in \mathbb{R}^{192 \times 192 \times 192}$ is split into $2^3$ non-overlapping patches, flattened, and projected to $C_0 = 48$ channels, yielding $96^3$ tokens. Successive Swin stages with $L = \{2, 2, 6, 2\}$ blocks halve the spatial resolution via patch-merge layers and grow the channel widths to $[48, 96, 192, 384]$. Windowed multi-head self-attention (W-MSA) and shifted-window MSA alternate within each block, providing $\mathcal{O}(N)$ computational complexity.

*2) Decoder and skip connections:* The decoder mirrors the encoder with patch-expanding layers that upsample the feature maps and concatenate the encoder activations. Each upsampling step is followed by two $3 \times 3 \times 3$ convolutions, group normalization, and GELU activation. The final head consists of a $1 \times 1 \times 1$ convolution that maps the feature volume to three logits (background, posterior leaflet, anterior leaflet).

*3) Temperature scaling for QA:* Neural networks are notoriously over-confident; uncalibrated soft-max probabilities distort the Shannon-entropy metric that underpins our reliability gate. Following Guo et al. [32], we apply a scalar *temperature* to the logits prior to soft-max: $\tilde{\mathbf{p}} = \text{softmax}(\mathbf{z}/T), \quad T > 1$, and tune $T$ on the validation set by minimizing the expected calibration error (ECE). The resulting Shannon-entropy map serves as the $95^{\text{th}}$-percentile QA score detailed in Section II-G.

### E. Training Strategy

The network is trained on the MVSeg-2023 training split ($N_{train} = 105$) and tuned on the validation split ($N_{val} = 30$). All experiments use mixed-precision (`torch.cuda.amp`) and are executed in a Google Colab Pro environment on a single NVIDIA A100 GPU with 80 GB on-board VRAM and 80 GB of system RAM, running `PyTorch 1.13` and `MONAI 1.3`. Table II lists the principal hyper-parameters.

*1) Data loading and augmentation:* Volumes are streamed through a caching `DataLoader` (`CacheDataset`, cache rate 1.0) to minimize I/O overhead. Online augmentation considers histogram shift ($p = 0.3$), Gaussian noise ($\sigma = 0.01$, $p = 0.3$), random left–right/antero–posterior flips ($p = 0.5$), and random $90°$ rotations ($p = 0.5$). Histogram shifting emulates operator-dependent gain settings, Gaussian noise mimics ultrasound speckle, and random flips/rotations compensate for variable probe orientation in trans-esophageal imaging. No augmentation is applied during validation or testing.

*2) Optimization policy.:* We employ AdamW with an initial learning rate of $3 \times 10^{-4}$, weight–decay $10^{-4}$, and a 500-iteration linear warm-up followed by cosine decay to $10^{-6}$. Gradients are clipped to a global norm of 1.0. Early stopping with a patience of 15 epochs selects the best checkpoint according to mean foreground Dice on the validation set.

*3) Loss Function:* To balance volumetric overlap with boundary fidelity, the network is optimized with a composite objective $\mathcal{L} = \lambda \mathcal{L}_{\text{Dice}} + (1-\lambda) \mathcal{L}_{\text{CE}}, \quad \lambda = 0.5$, where $\mathcal{L}_{\text{CE}}$

TABLE I

LAYER-BY-LAYER SPECIFICATION OF THE SWIN–UNETR CONFIGURATION USED IN THIS WORK. OUTPUT RESOLUTIONS ARE EXPRESSED RELATIVE TO THE INPUT SIZE $H \times W \times D = 192 \times 192 \times 192$. SWIN-UNETR CONTAINS $\approx 27.3$ M TRAINABLE PARAMETERS

| Stage | Output size | Patch / stride | Channels | Win. | $L$ | Main operations | Skip to |
|---|---|---|---|---|---|---|---|
| Patch embed | $\frac{1}{2}H \times \frac{1}{2}W \times \frac{1}{2}D$ | $2^3$ / 2 | $C_0$ | – | – | $2^3$ patch + linear proj. | – |
| **Encoder** | | | | | | | |
| Stage E1 | $\frac{1}{2}H \times \frac{1}{2}W \times \frac{1}{2}D$ | – | $C_0$ | $4^3$ | 2 | W-MSA → SW-MSA + MLP | Decoder D3 |
| Stage E2 | $\frac{1}{4}H \times \frac{1}{4}W \times \frac{1}{4}D$ | merge 2 | $2C_0$ | $4^3$ | 2 | Patch merge + Swin block | Decoder D2 |
| Stage E3 | $\frac{1}{8}H \times \frac{1}{8}W \times \frac{1}{8}D$ | merge 2 | $4C_0$ | $4^3$ | 6 | Patch merge + Swin block | Decoder D1 |
| Stage E4 | $\frac{1}{16}H \times \frac{1}{16}W \times \frac{1}{16}D$ | merge 2 | $8C_0$ | $4^3$ | 2 | Patch merge + Swin block | – |
| Bottleneck | $\frac{1}{16}H \times \frac{1}{16}W \times \frac{1}{16}D$ | – | $8C_0$ | – | – | $1 \times 1 \times 1$ conv + GELU | – |
| **Decoder** | | | | | | | |
| Stage D1 | $\frac{1}{8}H \times \frac{1}{8}W \times \frac{1}{8}D$ | expand 2 | $4C_0$ | – | – | Patch expand + 2×Conv ($3^3$) | – |
| Stage D2 | $\frac{1}{4}H \times \frac{1}{4}W \times \frac{1}{4}D$ | expand 2 | $2C_0$ | – | – | Patch expand + 2×Conv ($3^3$) | – |
| Stage D3 | $\frac{1}{2}H \times \frac{1}{2}W \times \frac{1}{2}D$ | expand 2 | $C_0$ | – | – | Patch expand + 2×Conv ($3^3$) | – |
| Output head | $H \times W \times D$ | upsample 2 | 3 | – | – | $1 \times 1 \times 1$ conv → soft-max | – |

TABLE II
KEY TRAINING HYPER-PARAMETERS.

| Category | Parameter | Value |
|---|---|---|
| Data | Resampled spacing | 0.6 mm isotropic |
| | Batch size ($B$) | 1 volume |
| | Train / val / test | 105 / 30 / 40 cases |
| Augmentation | Histogram shift | $p = 0.3$ |
| | Gaussian noise | $\sigma = 0.01$, $p = 0.3$ |
| | Random flip | $p = 0.5$ (LR or AP axis) |
| | Random 90° rotation | $p = 0.5$ (any axis) |
| Optimizer | Algorithm | AdamW |
| | Initial learning rate | $3 \times 10^{-4}$ |
| | Weight decay | $10^{-4}$ |
| | Warm-up iterations | 500 |
| | LR scheduler | Cosine → $10^{-6}$ |
| | Gradient clipping | Norm $\leq 1.0$ |
| Training loop | Max epochs | 120 |
| | Early-stop patience | 15 (val. Dice) |
| | Checkpoint metric | Mean foreground Dice |
| | Temperature $T$ | 1.9 |

is the multi-class cross-entropy and $\mathcal{L}_{\text{Dice}}$ is a soft Dice loss computed only for the anterior and posterior leaflet classes:

$$\mathcal{L}_{\text{Dice}} = 1 - \frac{2 \sum_{c=1}^{2} y_c \hat{y}_c + \varepsilon}{\sum_{c=1}^{2} y_c + \sum_{c=1}^{2} \hat{y}_c + \varepsilon}, \quad \varepsilon = 10^{-5} \quad (2)$$

The foreground-only Dice term mitigates the severe class imbalance introduced by the large background region, while the cross-entropy term penalizes thin-edge mis-registration.

### F. Evaluation Metrics

Segmentation accuracy are quantified with two complementary measures: the (class-averaged) Dice coefficient and the 95th-percentile Hausdorff distance (HD$_{95}$).

*1) Dice coefficient:* For each foreground class $c \in \{1, 2\}$ (posterior, anterior leaflet) the Dice index is calculated as $\text{Dice}_c = 2|P_c \cap G_c| / (|P_c| + |G_c|)$, where $P_c$ and $G_c$ denote the predicted and ground-truth voxel sets, respectively.

*2) HD$_{95}$:* The symmetric Hausdorff distance between the predicted surface $\mathcal{S}_P$ and the reference surface $\mathcal{S}_G$ is

$$\text{HD} = \max\Big\{ \sup_{\mathbf{p} \in \mathcal{S}_P} \inf_{\mathbf{g} \in \mathcal{S}_G} \|\mathbf{p} - \mathbf{g}\|_2, \sup_{\mathbf{g} \in \mathcal{S}_G} \inf_{\mathbf{p} \in \mathcal{S}_P} \|\mathbf{g} - \mathbf{p}\|_2 \Big\} \quad (3)$$

Because single outlier voxels can inflate the classical Hausdorff distance, we adopt the 95th-percentile variant, HD$_{95}$, which replaces the outer maxima with the 95th-percentile of the bidirectional point-wise distance distribution [33].

*3) Surface-specific metrics:* To quantify geometric fidelity of the exported STL surfaces, we report the average symmetric surface distance (ASSD) and the percentage of non-manifold edges (%NME). Let $\mathcal{S}_P$ and $\mathcal{S}_G$ denote the predicted and ground-truth meshes, and let $d(\mathbf{p}, \mathcal{S}) = \min_{\mathbf{g} \in \mathcal{S}} \|\mathbf{p} - \mathbf{g}\|_2$ be the point-to-surface distance. The ASSD is then

$$\text{ASSD} = \frac{1}{|\mathcal{S}_P| + |\mathcal{S}_G|} \Big( \sum_{\mathbf{p} \in \mathcal{S}_P} d(\mathbf{p}, \mathcal{S}_G) + \sum_{\mathbf{g} \in \mathcal{S}_G} d(\mathbf{g}, \mathcal{S}_P) \Big) \quad (4)$$

expressed in millimetres. Mesh watertightness is evaluated by counting edges incident to more than two triangular faces; the non-manifold rate is $\%\text{NME} = 100 \times N_{\text{non-manifold}} / N_{\text{edges}}$. Lower values indicate cleaner topology and superior suitability for finite-element or flow simulation workflows.

### G. Quality Assurance

Automatic leaflet segmentations must be accompanied by a confidence indicator before they can be trusted in the operating room. We therefore equip the network with a single–pass, entropy–based QA gate that accepts or rejects each prediction without resorting to ensembles or Monte–Carlo sampling.

*1) Voxel-level entropy map:* Let $\tilde{\mathbf{p}} \in \mathbb{R}^{3 \times H \times W \times D}$ denote the soft-max probabilities obtained after temperature scaling (Section II-D3). The voxel-wise Shannon entropy is $H(\mathbf{v}) = - \sum_{c=1}^{3} \tilde{p}_c(\mathbf{v}) \ln \tilde{p}_c(\mathbf{v})$, where $\mathbf{v}$ indexes spatial position. Higher entropy indicates greater class ambiguity.

*2) Case-level score:* For each TEE volume, we compute the 95th percentile of the entropy distribution, $H_{95} = \text{quantile}_{0.95}\big(H(\mathbf{v})\big)$, which is robust to small clusters of noisy voxels yet sensitive to globally uncertain predictions.

*3) Threshold selection:* A threshold $\tau = 0.80$ was chosen on the validation set by maximising the Youden index of the ROC curve that discriminates *good* (Dice $\geq 0.80$) from *poor* segmentations. This single scalar is subsequently applied to the unseen test set: Accept $\Leftrightarrow H_{95} < \tau$.

### H. Mesh Generation

The final stage converts the binary leaflet mask into a watertight STL surface suitable for computational hemodynamic, finite-element analysis, and augmented-reality display. The entire routine is implemented in `PyVista 0.40` with `CuPy` back-ends, achieving full GPU acceleration.

The predicted mask is first padded to a multiple of 32 voxels along every axis and then subjected to a two-iteration binary closing operation to eliminate isolated voids at the free edge. For GPU-accelerated marching cubes, we consider an isovalue of 0.5 and the voxel spacing $\Delta = (0.6, 0.6, 0.6)$ mm. The kernel produces a triangular surface $\mathcal{S}_0 = (\mathbf{V}, \mathbf{F})$.

Subsequently, we perform connected-component filtering, where all components are labeled with a breadth-first search on $\mathcal{S}_0$. Only the largest component is retained, which removed spurious facets per test case. To remove staircase artifacts while preserving edge length, we perform 30 iterations of Taubin $\lambda$–$\mu$ smoothing [29] with coefficients $\lambda = 0.5$ and $\mu = -0.53$. The smoothed mesh $\mathcal{S}_T$ is re-centered and rotated so that its long axis aligns with $\hat{\mathbf{e}}_z$ using a principal component analysis [34] on the vertex cloud. This yields consistent camera orientation for qualitative rendering. The final surface $\mathcal{S}_T$ is serialized as a binary STL via `pyvista.save()`.

## III. RESULTS AND DISCUSSION

### A. Swin-UNETR Training

The model training results are presented in Fig. 2. The composite loss decreased by 45 % within the first five epochs. Performance plateaued after around 50 epochs.

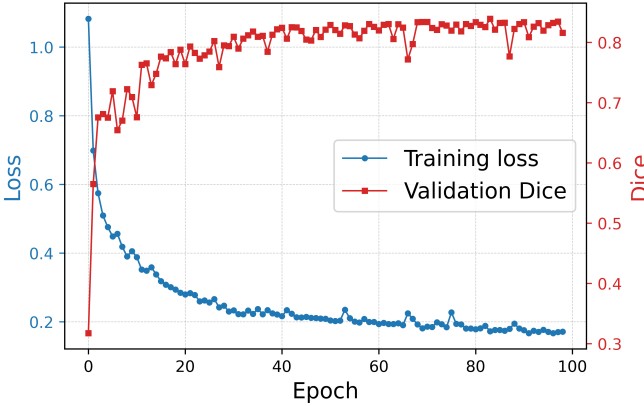

Fig. 2. Swin-UNETR learning curve. Composite training loss (blue, left axis) and validation Dice (red, right axis) over 100 epochs.

### B. Quantitative Segmentation Accuracy

The trained model was evaluated on the 40 withheld test cases of MVSeg-2023. The posterior + anterior leaflet Dice coefficient achieved a median of $0.832 \pm 0.051$, and the $HD_{95}$ measured $4.2 \pm 2.1$ mm with a single outlier at 25.6 mm. Compared with the official MVSeg-3DTEE challenge benchmark—DeepMitral [10], which reported a Dice score = 0.79 and $HD_{95}$ = 2.22 mm—our Swin-UNETR pipeline improved the segmentation overlap to a median Dice = 0.832 while delivering a clinically acceptable $HD_{95}$. Fig. 3 visualizes the full distribution of both metrics. The violin + box plots reveal a slight left skew in Dice and a heavy-tailed $HD_{95}$ distribution. Interquartile ranges were narrow (Dice IQR = 0.79–0.86, $HD_{95}$ IQR = 2.6–6.8 mm).

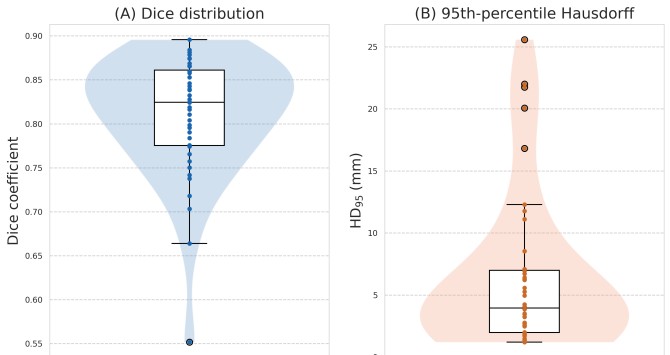

Fig. 3. Quantitative performance on the 40-case test set. **(A)** Dice coefficient distribution and **(B)** 95th-percentile Hausdorff distance ($HD_{95}$). Each dot represents one test TEE volume; violin envelopes show kernel densities, and embedded box plots indicate median and interquartile range.

### C. Quantitative Surface Analysis

Visual inspection confirmed that the network reproduced the global saddle shape and thin free edges of both mitral leaflets in the test cases. Fig. 4 contrasts two representative subjects. Surface-oriented analysis confirmed that the reconstructed meshes are both geometrically accurate and topologically clean. ASSD was computed to be $0.39 \pm 0.013$ mm, which is marginally better than the 0.38 mm ASSD reported for the DeepMitral benchmark. Mesh watertightness was likewise high: %NME was $0.21 \pm 0.07$%. These figures indicate that the reconstructed leaflets require no manual repair before downstream finite-element or computational-flow simulations.

### D. Reliability of the Entropy-Based QA Gate

The single-pass QA mechanism aimed to retain high-quality predictions while rejecting potentially unreliable masks. When the validation-derived threshold of , $\tau = 0.80$, was applied to the test set, all 40 cases (100%) were accepted, and no volumes were rejected as low-confidence. Within this cohort, $H_{95}$ remained uncorrelated with Dice ($r = 0.016$, $p = 0.92$), confirming that high-confidence predictions consistently aligned with high segmentation accuracy. Fig. 5 illustrates this lack of correlation, where each dot represents an accepted test volume, and the gray band denotes the 95% confidence interval.

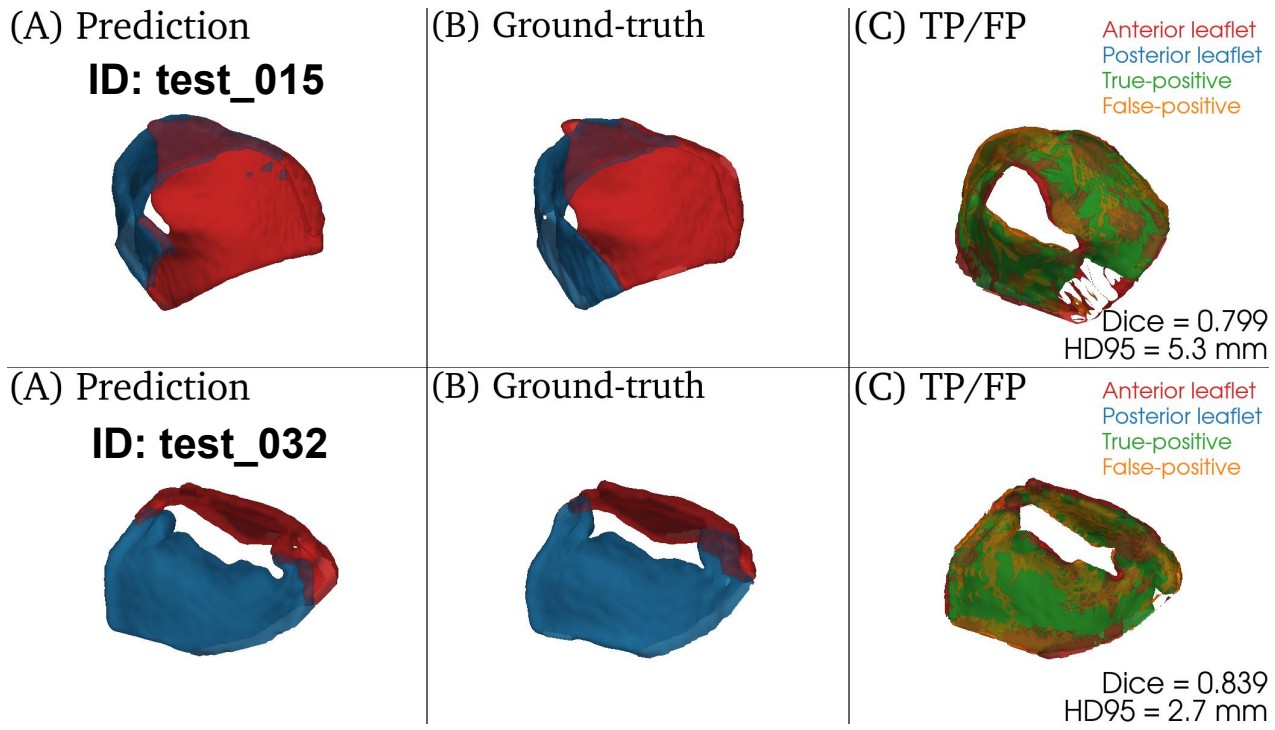

Fig. 4. Qualitative 3-D surface results for two test volumes. Columns show **(A)** network prediction, **(B)** ground-truth, and **(C)** voxel-level true-positive (TP) (green)/false-positive (FP) (orange) overlay. Dice and $HD_{95}$ are reported per case. Color code: anterior leaflet (red), posterior leaflet (blue).

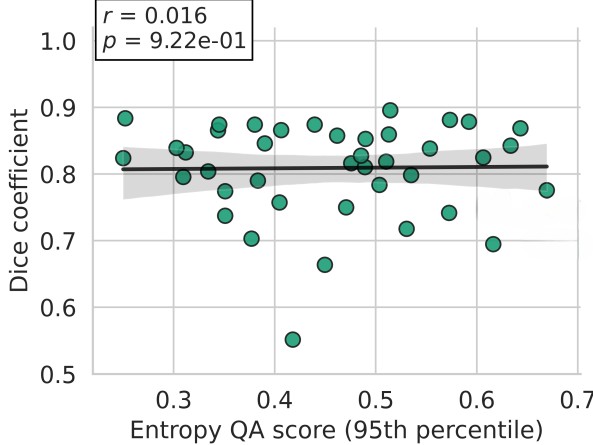

Fig. 5. Relationship between $H_{95}$ and Dice coefficient.

### E. Runtime and Memory Footprint

End-to-end inference—including segmentation, entropy-based QA, and mesh export—was benchmarked on the 40 hidden test volumes in the same Google Colab Pro environment used for training, i.e., a single NVIDIA A100 GPU with 40 GB VRAM and 83 GB system RAM. As summarized in Table III, the pipeline required 122 ms on average to process a $192^3$ volume, of which 104 ms were spent in the Swin-UNETR forward pass and 18 ms in the meshing routine. Peak GPU memory never exceeded 4.0 GB, leaving 90% of the A100's capacity available for batch processing or ensemble evaluation.

TABLE III
INFERENCE SPEED AND MEMORY USAGE ON 40 TEST VOLUMES (A100 40 GB, BATCH SIZE 1).

| Metric | Value |
|---|---|
| Segmentation latency (mean $\pm$ SD) | $104.4 \pm 34.3$ ms |
| Full-pipeline latency (mean $\pm$ SD) | $122.1 \pm 35.2$ ms |
| Throughput | 8.2 volumes s$^{-1}$ |
| Peak GPU memory | 4.0 GB of 40 GB |

### IV. CONCLUSION

This study presents the first end-to-end framework that unifies transformer-based voxel segmentation, single-pass entropy-driven QA, and sub-second mesh reconstruction for 3-D TEE of the MV. Several limitations must be acknowledged. First, the MVSeg-2023 dataset originates from a single institution and ultrasound vendor, so model generalizability to other scanners, probe types, and pathological subgroups remains to be established. Second, although region-overlap accuracy is state-of-the-art, the boundary error is still higher than the benchmark's error, suggesting that mesh-aware loss functions could further sharpen leaflet edges. Finally, the network operates on static end-diastolic frames and does not exploit temporal coherence across the cardiac cycle. Despite these caveats, the approach holds promise for intra-operative visualization, real-time computational hemodynamics, and the rapid morphological assessments required during MV surgery. Future work will broaden validation to multi-vendor, multi-

pathology datasets; extend the model to 4-D TEE sequences for temporally consistent segmentation; incorporate domain-adaptation and mesh-aware loss functions to further sharpen leaflet boundaries; and embed the entropy-based QA and instant STL export into an augmented-reality interface for prospective clinical evaluation during mitral interventions.

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
