# OpenReview forum: "3‑D TEE Mitral Valve Segmentation and Mesh Reconstruction with Real‑Time Quality Assurance"
_IEEE.org/EMBS/BHI/2025/Conference — BHI 2025_

### Official Review · Reviewer_43wr · 2025-07-07
**3-D TEE Mitral Valve Segmentation and Mesh Reconstruction with Real-Time Quality Assurance**

**Confidence:** 4
**Clarity Of Writing:** good
**Clinical Significance:** good
**Methodological Novelty:** good
**Overall Rating:** 7

**Experiments And Results:**

good

**Questions For The Authors:**

•	How can incorporating temporal coherence from 4-D TEE sequences enhance the accuracy and clinical utility of transformer-based MV segmentation models compared to the current static-frame approach?

**Strengths:**

•	Despite some caveats, the approach seems to hold promise for intra-operative visualization, real-time computational hemodynamics, and the rapid morphological assessments required during MV surgery.

**Summary Of The Paper:**

•	This study seems to present the first end-to-end framework that unifies transformer-based voxel segmentation, single-pass entropy-driven QA, and sub-second mesh reconstruction for 3-D TEE of the MV.

**Weaknesses:**

•	I did not come across any weakness.

---

### Official Review · Reviewer_BwkE · 2025-07-11
**3‑D TEE Mitral Valve Segmentation and Mesh Reconstruction with Real‑Time Quality Assurance**

**Confidence:** 5
**Clarity Of Writing:** great
**Clinical Significance:** great
**Methodological Novelty:** excellent
**Overall Rating:** 7

**Experiments And Results:**

good

**Questions For The Authors:**

How does the runtime and performance scale with larger batch sizes or higher-resolution inputs?

**Strengths:**

Trained and tested on the MVSeg-2023 dataset, the method achieves strong segmentation performance (Dice coefficient of 0.83 ± 0.05) with sub-second inference time, offering significant clinical utility for intra-operative guidance. A notable contribution is the use of a single-pass entropy score to assess segmentation quality without the computational burden of ensemble methods.

**Summary Of The Paper:**

This paper introduces a novel real-time pipeline for segmenting and reconstructing the mitral valve (MV) from 3-D transesophageal echocardiography (TEE) using a vision transformer architecture. The approach integrates voxel-wise segmentation via Swin-UNETR, entropy-based quality assurance for prediction reliability, and GPU-accelerated mesh reconstruction.

**Weaknesses:**

The method is technically innovative, clinically relevant, and well-optimized for real-time deployment. However, it also presents some limitations. The dataset used is from a single institution and vendor, which may restrict generalizability, as reported by the authors. The system currently only supports static end-diastolic frames and does not leverage the temporal dimension inherent in 4-D TEE imaging. Furthermore, while the QA strategy is promising, no baseline comparisons or ablation studies are provided to quantify its added value. Although segmentation accuracy is high, boundary precision still lags behind ideal levels.

---

### Official Review · Reviewer_kMKh · 2025-07-15
**3-D TEE Mitral Valve Segmentation and Mesh Reconstruction with Real-Time Quality Assurance**

**Confidence:** 4
**Clarity Of Writing:** great
**Clinical Significance:** great
**Methodological Novelty:** good
**Overall Rating:** 7

**Experiments And Results:**

good

**Questions For The Authors:**

What about other ultrasound vendors? Since this was all trained and tested on Philips systems, do you expect the same performance on GE or Siemens? Would domain adaptation be needed?

Could mesh-aware loss functions (e.g., Chamfer distance) further reduce HD95 without compromising speed?

How about making edges sharper? Have you looked into boundary-aware loss functions (like Hausdorff or surface loss) to better capture the thin leaflet edges?

Any clinical trials in sight? How close is this pipeline to being embedded into live TEE systems in the OR? What sort of validation do you see as the next step?

**Strengths:**

This work effectively integrates multiple innovative concepts, leveraging advanced transformer architectures, incorporating uncertainty estimation through entropy, and enabling efficient mesh export into a cohesive and practical pipeline.

It processes everything in about 100 ms, which is impressive and shows real potential for real-time use in the OR.

The entropy-based QA is simple yet effective, providing a useful safeguard without heavy computational costs.

This tackles a well-known bottleneck, speeding up mitral valve assessment during procedures, which could genuinely help clinicians.

They go beyond just Dice scores to also look at surface distances and mesh quality, which matters for simulations or AR overlays.

**Summary Of The Paper:**

The paper presents a vision-transformer-based pipeline for segmenting the mitral valve (MV) from 3-D transesophageal echocardiography (TEE) images, incorporating real-time quality assurance (QA) and mesh reconstruction. The proposed method uses a Swin-UNETR backbone trained on the MVSeg2023 dataset, achieving a Dice coefficient of 0.83±0.05 and a 95th-percentile Hausdorff distance of 4.2 mm. Key innovations include entropy-driven QA to flag uncertain predictions and GPU-accelerated mesh generation (104 ms per volume). The authors position their work as the first to unify transformer-based segmentation, uncertainty-aware QA, and real-time mesh reconstruction for intra-procedural applications.

**Weaknesses:**

Generalizability: Limited to single-institution, single-vendor data; performance on diverse scanners/pathologies is untested.

Boundary details still tricky: Even though overall overlap is strong, the thin leaflet edges still show errors around 4 mm (HD95), which could matter for precise surgical planning.

QA Validation: All test cases passed QA (H95 <0.80), raising questions about the threshold’s discriminative power in noisier scenarios.